

# A taste for exotic food: Neotropical land planarians feeding on an invasive flatworm

Piter K. Boll, Ilana Rossi, Silvana V. Amaral and Ana Leal-Zanchet

Instituto de Pesquisas de Planárias e Programa de Pós-Graduação em Biologia, Universidade do Vale do Rio dos Sinos—UNISINOS, São Leopoldo, Rio Grande do Sul, Brazil

## ABSTRACT

Invasive species establish successfully in new habitats especially due to their ability to include new species in their diet and due to the freedom from natural enemies. However, native species may also adapt to the use of new elements in their ecosystem. The planarian *Endeavouria septemlineata*, first recorded in Hawaii, was later found in Brazil. Recently, we found it in human-disturbed areas in southern Brazil and here we investigate its interactions with other invertebrates both in the field and in the laboratory. We observed the species in the field during collecting activities and hence maintained some specimens alive in small terraria in the laboratory, where we offered different invertebrate species as potential prey and also put them in contact with native land planarians in order to examine their interaction. Both in the field and in the laboratory, *E. septemlineata* showed a gregarious behavior and was found feeding on woodlice, millipedes, earwigs and gastropods. In the laboratory, specimens often did not attack live prey, but immediately approached dead specimens, indicating a scavenging behavior. In an experiment using the slug *Deroceras laeve* and the woodlouse *Atlantoscia floridana,* there was a higher consumption of dead specimens of woodlice and slugs compared to live specimens, as well as a higher consumption of dead woodlice over dead slugs. Four native land planarians of the genus *Obama* and one of the genus *Paraba* attacked and consumed *E. septemlineata*, which, after the beginning of the attack, tried to escape by tumbling or using autotomy. As a scavenger, *E. septemlineata* would have no impact on the populations of species used as food, but could possibly exclude native scavengers by competition. On the other hand, its consumption by native land planarians may control its spread and thus reduce its impact on the ecosystem.

## INTRODUCTION

Most invasive species establish successfully in new habitats because of their ability to adapt their diets to include new species and due to the freedom from natural enemies (*Colautti et al., 2004*; *Prior et al., 2014*). However, as easily as an introduced species may adapt to a new environment, native species may adapt to a new element in their ecosystem. For example, native predators often adapt to exotic prey, controlling their populations (*Carlsson, Sarnelle & Strayer, 2009*). Conversely, an invasive predator may benefit from the

Corresponding author
Ana Leal-Zanchet,
zanchet@unisinos.br

ineffective antipredator behavior of native species, thus increasing its chances of capturing prey (*Sih et al., 2010*).

Land planarians are important predators of other invertebrates in the soil fauna in tropical regions (*Ogren, 1995*; *Sluys, 1999*). Despite being sensitive to dehydration, high temperatures and luminosity, some species are able to adapt to human-disturbed environments and become invasive when colonizing areas outside their native range (*Froehlich, 1955*). The success of most invasive land planarians is attributed to their feeding habits, which gives them the ability to adapt their diet to include local invertebrate species, thus leading to a quick dispersal (*Murchie et al., 2003*). Furthermore, the naïveté of native species to respond effectively to predation has been shown to increase the predation success of planarians (*Fiore et al., 2004*). The possibility of natural enemies to control invasive planarians is usually not taken into account because it is assumed that land planarians are top predators and therefore lack predators (*Sluys, 1999*).

Currently, several land planarians of the subfamily Rhynchodeminae that are native to Australasia are known as important invasive species (*Winsor, Johns & Barker, 2004*; *Justine, Thévenot & Winsor, 2014*; *Álvarez-Presas et al., 2014*; *Justine et al., 2015*). In Hawaii, *Mead (1963)* found that a rhynchodeminid planarian from tribe Caenoplanini, *Endeavouria septemlineata* (*Hyman, 1939*), is an effective predator on the introduced giant African snail. He also observed the species feeding on earthworms and small insects. Later, *E. semptemlineata* was recorded for Brazil, although the impact of its invasion over native species was not studied (*Carbayo, Pedroni & Froehlich, 2008*).

Recently, we found *E. septemlineata* in human-disturbed areas in southern Brazil. In order to understand its impact on the ecosystem and its effectiveness as an invasive species, we investigated its interactions with other invertebrates, both in the field and in the laboratory. Our predictions are that, as an effective invader, *E. septemlineata* would feed on a wide range of invertebrates. We also investigated whether or not native land planarians may use *E. septemlineata* as an alternative food source.

## MATERIAL AND METHODS

We found specimens of *E. semptelineata* in gardens in Montenegro (29°40′S, 51°28′W) and Campo Bom (29°40′S, 51°3′W), and in the Porto Alegre Botanical Garden (30°03′S, 51°10′W), in Porto Alegre, Brazil. We documented occasional observations related to behavior in the field during collecting activities in the city of Montenegro and captured several specimens, taking them to the laboratory. We placed the planarians in plastic terraria with moist soil, leaves and log fragments at a temperature of 20 °C and a relative air humidity of 90%.

We killed nine specimens in hot water and fixed them in 10% buffered formalin. Later, we processed these specimens histologically (following *Froehlich & Leal-Zanchet, 2003*) for taxonomic identification through examination of the internal morphology and deposited them in the reference collection of Museu de Zoologia da Universidade do Vale do Rio dos Sinos, São Leopoldo, Rio Grande do Sul, Brazil.

In the laboratory, we kept three groups of several specimens of *E. septemlineata* (between 10 and 30 individuals) alive in terraria with various invertebrate species as potential food items: gastropods *Deroceras laeve* (OF Müller, 1774) and *Bradybaena similaris* (Férussac, 1821); isopods *Atlantoscia floridana* (Van Name, 1940), *Benthana cairensis* Skolowicz, Araujo & Boelter, 2008, *Porcellio scaber* Latreille, 1804 and *Armadillidium vulgare* Latreille, 1804; termites *Nasutitermes* sp.; diplopods *Rhinocricus* sp.; and earthworms *Eisenia andrei* Bouché, 1972 and *Metaphire schmardae* (Horst 1883). We examined the terraria twice a week, searching for planarians feeding on the invertebrates.

Later, we performed two experiments of interaction. In the first one, we kept nine groups of five individuals of *E. septemlineata* alive in small terraria and offered two different invertebrates as food, the slug *D. leave* and the woodlouse *A. floridana*, in either of two states, alive or dead, totaling four treatments: live *A. floridana*, dead *A. floridana*, live *D. laeve* and dead *D. laeve*. We put five specimens of one of the treatments with each group of *E. septemlineata* for 24 h. After that time, we recorded the number of items consumed and removed all food, putting the planarians in fast for 48 h before offering a different food of random choice. We offered each treatment ten times throughout the nine groups and compared the mean number of consumed prey items in each treatment using a one-way analysis of variance with an *a posteriori* Tukey test.

In the second experiment, we tested the possibility of predation on *E. septemlineata* by native predators. Most known predators of land planarians in the Neotropical ecozone are other land planarians (*Froehlich, 1955*), especially from the genera *Obama* and *Paraba* (*Froehlich, 1955*; *Hauser & Maurmann, 1959*). Therefore, we offered *E. semptemlineata* to native species of land planarians of these two genera, viz., *Obama anthropophila* Amaral, Leal-Zanchet & Carbayo, 2015, *O. carrierei* (Graff, 1897), *O. ficki* (Amaral & Leal-Zanchet, 2012), *O. josefi* (Carbayo & Leal-Zanchet, 2001), *O. ladislavii* (Graff, 1899), *O. marmorata* (Schultze & Müller, 1857) and *Paraba multicolor* (Graff, 1899). We put one specimen of the native planarian and one specimen of *E. septemlineata* in a moistened Petri dish under low diffuse light and examined their interaction. We offered *E. septemlineata* fifteen times to each native land planarian species and compared the rate of attacks of each species on *E. septemlineata* using a chi-square test.

The work was conducted under the collection permits and licenses granted by Instituto Chico Mendes de Conservação da Biodiversidade (permit numbers 24357 and 26683).

## RESULTS

Both in the field and in the laboratory, specimens of *E. septemlineata* showed a gregarious behavior, constantly gathering in groups ranging from a few to tens of individuals (Fig. 1A). Specimens were often found in the field feeding in groups on *B. similaris* (Fig. 1B), *D. leave*, *A. vulgare* (Fig. 1C), *Rhinocricus* sp. (Fig. 1D), and earwigs, or individually on *A. vulgare* and *Nasutitermes* sp. (I Rossi, pers. obs., 2014).

In the laboratory, when in groups of 10–20 specimens, we found the planarians consuming the woodlice *A. vulgare* and *A. floridana* individually and the land snail

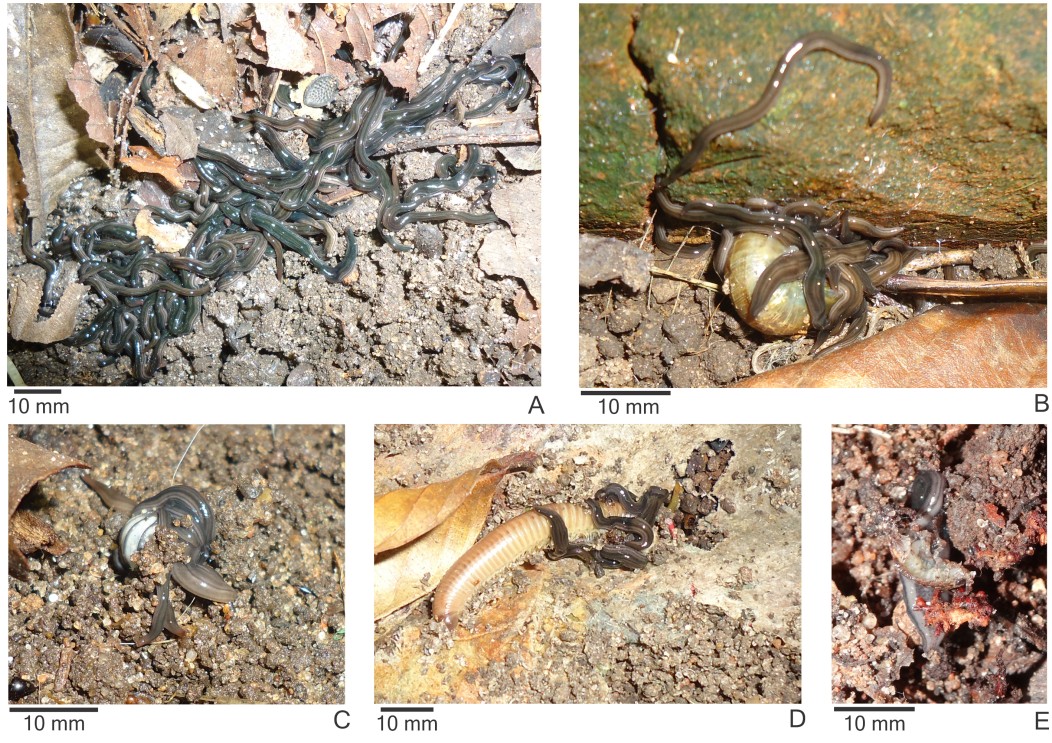

**Figure 1 Behavior of *Endeavouria septemlineata*.** (A) Specimens gathering in a group of many individuals in the field; specimens feeding on (B) the snail *Bradybaena similaris*, (C) the woodlouse *Armadillidium vulgare* and (D) the millipede *Rhinocricus* sp. in the field; (E) a single specimen feeding on the woodlouse *Atlantoscia floridana* in the laboratory.

*B. similaris* in groups. Despite being maintained in large groups with the invertebrates in the terraria for several days, specimens of *E. semptemlineata* were rarely found eating.

In the first experiment of interaction, *E. septemlineata* showed higher consumption of dead specimens over live specimens of both offered species. Also, there was a higher consumption of dead *A. floridana* over dead *D. laeve* (ANOVA, $F = 45.429$, $df = 3$, $p < 0.001$; Fig. 2).

Specimens of the native land planarians *O. josefi* (Fig. 3A), *O. marmorata* (Fig. 3B), *O. carrierei*, *O. anthropophila* (Fig. 3C) and *Paraba multicolor* (Fig. 3D) reacted to the encounter with *E. septemlineata* by capturing and consuming it. All species attacked and consumed *E. septemlineata* in more than 70% of the encounters, except *O. marmorata*, which only attacked in 53% of encounters ($\chi^2 = 16.337$; $df = 4$; $p = 0.003$).

At the beginning of the attack by the native planarians, individuals of *E. septemlineata* reacted by moving quickly away from the predator. The most frequent strategies to avoid predation were tumbling or, if the posterior end was trapped, autotomy of the posterior end. The planarian performed the tumbling behavior by lifting the posterior end and bending it forward until touching the substrate ahead of the anterior end (Fig. 4).

When in contact with native land planarians, individuals of *E. septemlineata* constantly showed a non-aggressive approaching behavior, often crawling either onto the dorsum or to the side of the individual of the other species and entering a resting position in a way

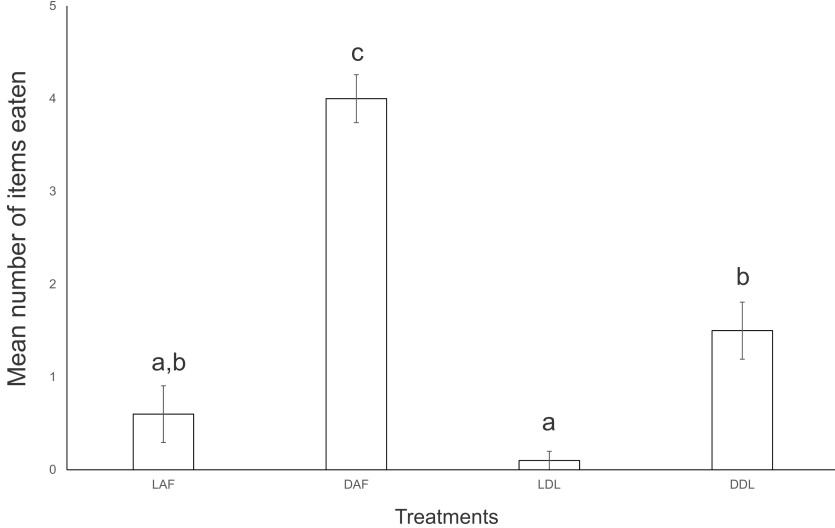

**Figure 2 Mean number of food items consumed by five specimens of *Endeavouria septemlineata* in 24 h.** LAF, live *Atlantoscia floridana*; DAF, dead *A. floridana*; LDL, live *Deroceras laeve*; DDL, dead *D. laeve*. Different letters indicate significant differences.

similar to the one showed towards conspecific individuals. They only started an escape response after the native planarian initiated the attack.

## DISCUSSION

We observed specimens of *E. septemlineata* from southern Brazil feeding on a great range of invertebrates, including mollusks and arthropods. In Hawaii, the invasive giant African snail became a greatly available food source for *E. septemlineata* (*Mead, 1963*). This led to a boost in the population and, consequently, the predation impact over native land snails increased. According to *Mead (1963)*, the low density of native land snails on the island makes them unlikely to be the main prey of the planarian. Our results corroborate this hypothesis of a diet including more than only gastropods. Such feeding habits, including both gastropods and arthropods, have been reported for other species of Caenoplanini, such as *Caenoplana coerulea* and *Parakontikia ventrolineata* (*Winsor, Johns & Barker, 2004*; *Breugelmans et al., 2012*), the latter also known to feed in groups (*Barker, 1989*).

Despite the fact that we find groups of *E. septemlineata* feeding on several invertebrates in the field, the specimens rarely eat live prey in the laboratory, frequently ignoring them. *Mead (1963)* reports feeding observations in the laboratory that included the capture of live snails, but gives no details about the conditions in which the planarians were maintained. *Carbayo, Pedroni & Froehlich (2008)* reported that, under laboratory conditions, *E. septemlineata* accepted snails and smashed slugs, but they also did not present details about the observations. Our results contrast with those observations, as there was higher consumption of dead woodlice in relation to dead or live slugs, as well as no statistical differences among the consumption of live or dead slugs and live woodlice.

In experiments with native species of *Microplana* in the United Kingdom, *McDonald & Jones (2007)* found that the planarians eat live prey less frequently in the laboratory

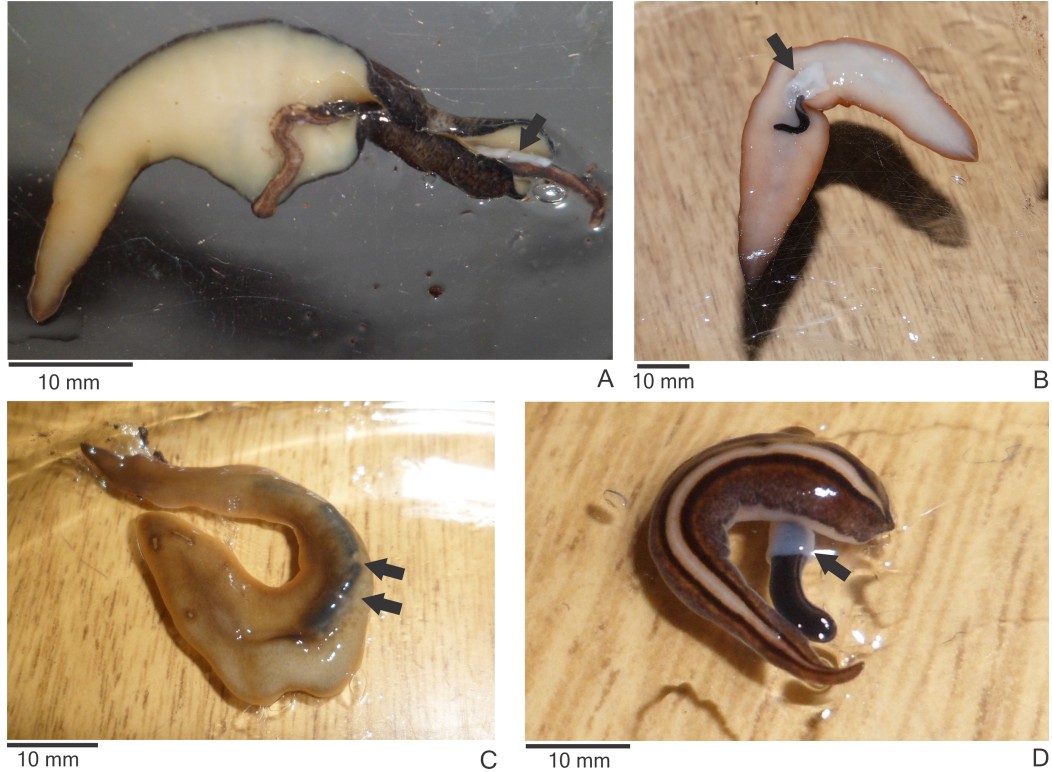

**Figure 3 Native land planarians consuming *Endeavouria septemlineata* in experiments in the laboratory.** (A) *Obama josefi* in ventral view; (B) *Obama marmorata* in ventral view; (C) *Obama anthropophila* in dorsal view; and (D) *Paraba multicolor* in dorsal view. Arrows indicate the pharynx of the predators; double arrows show parts of the body of a preyed specimen of *E. septemlineata* in the intestine.

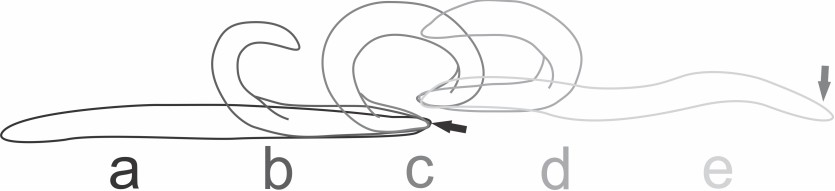

**Figure 4 Tumbling behavior of *Endeavouria septemlineata*.** (A) Initial position; (B) posterior end lifted and bent forward; (C) posterior end touching the substrate ahead of the anterior end; (D) anterior end lifted; (E) final position. Arrows show head in initial and final positions.

than in the field, although they accepted dead animals rather well. This may indicate that those species are more scavengers than predators. A species closely related to *E. septemlineata* that shows a similar gregarious behavior is *P. ventrolineata*. It has been reported to attack live prey actively in the field (*Barker, 1989*), but recent observations also indicate a scavenging behavior (*Justine, Thévenot & Winsor, 2014*). Our observations suggest that *E. semptemlineata* is mainly a scavenger.

As an obligate or facultative scavenger, *E. septemlineata* would have little effect on the population of most invertebrates it consumes as food, since it would not decrease

significantly the population size of those species. Nevertheless, it would decrease the amount of dead material available for native scavengers, leading to competition and, if having advantage over the resources, the invasive species may dislodge a native scavenger or force it to change its diet (*Wilson & Wolkovich, 2011*).

Nevertheless, despite the considerably wide occurrence of *E. septemlineata* in Brazil (*Carbayo, Pedroni & Froehlich, 2008*), its impact over the populations of native species may be under control due to its predation by native land planarians. In our study, which was restricted to southern Brazil, five native land planarians fed on *E. septemlineata*. When considering the range of distribution of this invasive species, it is likely that more predator species exist.

The control of invasive land planarians by predators has always seemed unlikely to succeed, as few natural predators are known (*Justine et al., 2014*). Besides beetles (*Gibson, Cosens & Buchanan, 1997*) and one snail species (*Lemos, Canello & Leal-Zanchet, 2012*), no other organisms have been reported to feed on introduced land planarians prior to the present work. Vertebrates usually do not accept land planarians as food as they seem to find them unpalatable (*Ducey et al., 1999*), although some birds may readily eat flatworms (H Jones, pers. comm., 2015).

Europe is the continent most affected by invasive land planarians (*Álvarez-Presas et al., 2014*; *Justine, Thévenot & Winsor, 2014*), but it also has a very small number of native species. Thus, native European animals are unlikely to predate land planarians, since land planarians compose a rare group on this continent and therefore are not available as a significant food resource. On the other hand, South America has a high richness of land planarians and effective predators are very likely to exist. One such predator, the land snail *Rectartemon depressus*, which consumes various native species of land flatworms, has been identified recently (*Lemos, Canello & Leal-Zanchet, 2012*). The consumption of the invasive *E. septemlineata* by native land planarians, including species common to urban environments, may be an important factor in controlling the dispersal of introduced land planarians in South America.

The inclusion of exotic prey in the diet of a native predator, sometimes even leading it to switch from a native to an exotic species as a main food source, is not uncommon (*Carlsson, Sarnelle & Strayer, 2009*), although it seems to vary considerably depending on the type of ecosystem and the trophic level (*Prior et al., 2014*). *Obama anthropophila*, *O. carrierei* and *O. josefi* seem to have other native land planarians as their main prey (P Boll, pers. obs., 2014) and thus may recognize *E. septemlineata* as a suitable species to replace native prey. The consequences of such interaction over the populations of both predator and prey depend on the responsive capacities of both species, including rapid adaptive change of individuals by learning or changes in morphology and behavior within a population due to natural selection (*Carlsson, Sarnelle & Strayer, 2009*).

Our results suggest that, as primarily a scavenger, *E. septemlineata* may not have significant effects on native species on which it feeds, and its spread may be controlled by native predators. However, it is possible that its presence significantly affects the trophic

web structure by dislodging native scavengers or altering the predation pressure on native preys by native predators.

## ACKNOWLEDGEMENTS

We thank Elisa von Groll, João Braccini, Lucas Schvambach, Márcio Sasamori and Victor Sawa for helping in the capture of land planarians; Ignacio Agudo, Marie Bartz and Patrícia Rodrigues for the help in identifying invertebrate species. We acknowledge MSc. Edward Benya for the English review of the text. Dr. Hugh Jones, Dr. Leigh Winsor and an anonymous reviewer are acknowledged for their constructive comments on an earlier draft of the paper.

### Funding

The Conselho Nacional de Desenvolvimento Científico e Tecnológico (CNPq), the Coordenação de Aperfeiçoamento de Pessoal de Nível Superior (CAPES) and the Fundação de Amparo à Pesquisa do Rio Grande do Sul (FAPERGS) provided research grants and fellowships in support of this study. The funders had no role in study design, data collection and analysis, decision to publish, or preparation of the manuscript.

### Grant Disclosures

The following grant information was disclosed by the authors:
Conselho Nacional de Desenvolvimento Científico e Tecnológico (CNPq).
Coordenação de Aperfeiçoamento de Pessoal de Nível Superior (CAPES).
Fundação de Amparo à Pesquisa do Rio Grande do Sul (FAPERGS).

### Competing Interests

The authors declare there are no competing interests.

### Author Contributions

- Piter K. Boll conceived and designed the experiments, performed the experiments, analyzed the data, wrote the paper, prepared figures and/or tables, reviewed drafts of the paper.
- Ilana Rossi performed the experiments, analyzed the data, prepared figures and/or tables, reviewed drafts of the paper, histological processing and analysis of the morphology for the identification of the flatworms.
- Silvana V. Amaral performed the experiments, analyzed the data, reviewed drafts of the paper, histological processing and analysis of the morphology for the identification of the flatworms.
- Ana Leal-Zanchet analyzed the data, contributed reagents/materials/analysis tools, wrote the paper, reviewed drafts of the paper, analysis of the morphology for the identification of the flatworms.

## Field Study Permissions

The following information was supplied relating to field study approvals (i.e., approving body and any reference numbers):

The work was conducted under the collection permits and licenses granted by Instituto Chico Mendes de Conservação da Biodiversidade (permit numbers 24357 and 26683).

## Supplemental Information

Supplemental information for this article can be found online at http://dx.doi.org/10.7717/peerj.1307#supplemental-information.

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
