# Peer review of "A taste for exotic food: Neotropical land planarians feeding on an invasive flatworm"

_PeerJ, doi:10.7717/peerj.1307_

## Round 0.1 · original submission · Major Revisions

The reviewers considered that there are major flaws in the report of the experimental design. I believe that you need not only to edit the manuscript but also to perform additional laboratory experiments with a more precise protocol and significant numbers of observations.

·

Basic reporting

Error of fact: Abstract line 4, and text lines 56/57.
Contrary to what is stated in this manuscript based on information in another paper, currently there are no records of E. septemlineata in Australia, and I am unaware of any formal or informal reports or publications of the species in Australia to date.
The mention of E. septemlineata in Australia by Carbayo et al is erroneous. The presence of E. septemlineata in Australia was not reported in either of the two papers they cite (Fiore et al 2004 and Greenslade et al 2007) though other austral species are mentioned.
The abstract must be corrected, and the sentence on lines 56 and 57 deleted.

Experimental design

No comments

Validity of the findings

No comments

Additional comments

This is a most interesting and welcome paper concerning field observations of an introduced land planarian in Brazil, Endeavouria septemlineata its interactions in simple laboratory expertiments with potential prey and predators, and the possible significance of these findings.

·

Basic reporting

Boll et al. review.

Page 1.
Line 1. I disagree that most invasive species are generalist. Some may be but others can be very specific, e.g. Arthurdendyus triangulatus (the New Zealand flatworm) and Bipalium kewense feed exclusively on earthworms. Platydemus manokwari is a specialist molluscivore.
Line 2. Suggest replace “release” with “freedom”.
Line 4. Replace “registered” with “found”.

Page 2.
See comments for page 1.

General comment: it is usual to give the full name of every species in the paper, including the naming authority and date, at its first mention in the paper. In this case all the flatworms and all the actual or potential prey species.

Page 4.
Line 31-32. See above comments.
Line 49. “….native of Australasia….”.

Page 5.
Line 53. “….predator on the introduced…..”.
Line 59. “….. its impact on the ecosystem….”.
Line 70. Replace “leafs” with “leaves”.
Line 71. “….at a temperature….”.

Page 6.
Lines 75-76. Specimen numbers required.
Line 77. Delete comma after laboratory.

Page 7.
Line 104/5.”….were rarely found feeding” is the same as “……feeding was not very common”. But how common or how rare was it? Out of all the experiments, how many (either %, or proportion) ended with feeding? What was the prey?

This leads to general comments about the results: Much more detail is needed.
It would be useful to include some quantification. No mention is made of the results of offering two potential prey species in each experiment. What were the results and did the worms show any preference for one prey species over another? Was this done with live prey or dead/wounded prey? Any preference for dead/wounded over live? Is there any evidence that the worms show a preference for any particular prey species, or avoid a prey species?
What species were consumed alive?
What species were consumed when dead or wounded?
Were these single worms or many worms?

Line 111. Suggest replacing “showed an escaping behavior” with “reacted”.

Page 8.
Line 134. ….”frequently ignoring prey species.”

Page 9.
Line 148. “… the amount of dead material…”.
Line 152. Replace “this” by “the’.
Line 161. I know that birds will readily eat land flatworms (that is one reason they seek shelter during the day), though I am not aware of any publication.

References
I have not checked in full, but Santoro & Jones (2001) is in the list, but not mentioned in the text.

Figures.
The legends need to be clarified.
Figure 1. Legend needs re-writing. No need to repeat the name Endeavouria septemlineata. Describe each figure one at a time A to E – why have (B-D) followed by B etc. What is the prey in (D)?


Figure 3. What arrows?

Experimental design

See above

Validity of the findings

See above

Additional comments

I feel the results section needs enlarging to give more details of the numbers of feeding events in relation to the number of experiments (it is not enough to say that they were not common).

Reviewer 3 ·

Basic reporting

The ms. Studies the feeding preferences of an alien terrestrial planarian species introduced in Brazil, and also tests its possible predators. The results are interesting. However, I find that the experimental design and the results are poorly explained.

Experimental design

For the experimental design I find a lot of information missing. The autors explain that they ketp groups of 10 to 20 individuals of E. septemlineata in terraria, but they do not say how many such groups did they study. They neither explain how many invertebrates of each species did they offer as food in total to each group. They neither give a hint of how many were offered alive and how many dead.
In line 101 says “we found the planarians consuming the woodlice A. vulgare and A. floridana individually”, I wonder if this means that although being in a terraria with other 10 to 20 E. septemlineata the animals that fed on this woodlice did it individually or if this means that they only fed on this animals when one E. septemlineata was set alone with them (but this is not explained in the methods section, so I suppose the first option is the correct).

Validity of the findings

In the results section they explain the animals did not eat much, and it seems they preferred dead animals. But they do not give a single number, which I will have expected. From my point of view it is necessary that the authors include a table indicating how many individuals dead and how many alive from each invertebrate species were offered to each group of E. septemlineata, and how many were eaten alive and how many dead. I understand that when describing feeding of animals on the field one can simply explain punctual observations, but if an experiment has been run, the experiment have to have a clear experimental design and the numbers of the results must be given, not simply an statement of “feeding was not very common”, it is better something like from 15 animals offered 0 were eaten. The table can also include information on whether the animals fed in groups or alone.
For the second experiment it is the same, they do not explain how many individuals of each species were tested to eat E. septemlineata. I can understand that in this case the authors consider that if once one species has eaten the alien planarian is enough to demonstrate that the alien planarian can be considered a prey for it, but I think they may anyway inform in the paper on how many such interactions did they try and in how many cases there was predation.
The validity of the findings would be better judged when we have all these numbers, if not we only have a subjective appreciation of the authors.

Additional comments

A part from the need to include the numbers for the experimental design and results as I state in the previous section. In general, I also find some parts of the ms a little repetitive or lacking order (specially in the introduction and discussion).

---

## Round 0.2 · Minor Revisions

This manuscript is much better than the first version and I believe it can be transformed into a publishable manuscript with just some additional work.

(1) Reviewer 2 made a comment about the two first sentences of the text. This opinion might be discussed.

(2) Reviewer 3 made comments about the new parts of the text. Please consider these comments.

(3) You might add a reference to the recent paper by Justine et al. about the presence of Platydemus manokwari in the USA (PeerJ). I feel uncomfortable in suggesting to cite my own paper, and it is your choice of course, but this was published after you submitted your first version and you might have missed it.

·

Basic reporting

Much improved over the first version. Quantitative information of the results are now included.

Experimental design

No comments.

Validity of the findings

No comments.

Additional comments

I suggest the first two sentences of the Abstract are removed.
Perhaps too much is made of the invasive issue, but does not detract from the paper.
Re Page 10, line 174. I know from personal observations that birds (ducks) will readily eat flatworms.

Reviewer 3 ·

Basic reporting

The new version of the ms is much better than the previous one, the autors have redone some experiments and done a statistical analysis that allow them yielding objective results, which give answer to my initial review concerns. So, I think now the ms is ready to be accepted although there are some minor changes needed in the new parts added:
Abstract. The following sentence is lacking a “not”:
As a scavenger, E. septemlineata would NOT impact the populations of species used as food, but could possibly exclude native scavengers by competition

Line 117. They only refer to the statistical analysis applied by a reference to figure 2, then in figure 2 it says “Different letters indicate significant differences”. I think this is not clear enough, it would be better if in the main text they explain that they have found significant differences between the amount of live and dead animals of the same species eaten by the terrestrial planarians, which I think is the important result. If I understand well they find no statistical difference between LAF and LDL ("a" in both cases), neither between LAF and DDL ("b" in both cases, a surprising result). But, I do not think this comparison may worry them. It is also interesting the fact that terrestrial planarians seem to prefer DAF to DDL, since here there is also a statistical difference ("c" and "b"), so, the authors may also want to refer to that fact in their results or discussion.

Experimental design

The new experiments performed and their explanation in the ms are now addequate

Validity of the findings

No comments

---

## Round 0.3 · accepted · Accept

The corrections on the second revised version are satisfactory. No further comments!